# Genetic Diversity and Molecular Evolution of Hepatitis E Virus Within the Genus *Chirohepevirus* in Bats

**DOI:** 10.3390/v17030339

**Published:** 2025-02-28

**Authors:** Bo Wang, Peter Cronin, Marcus G. Mah, Xing-Lou Yang, Yvonne C. F. Su

**Affiliations:** 1Programme in Emerging Infectious Diseases, Duke-NUS Medical School, 8 College Road, Singapore 169857, Singapore; peter@duke-nus.edu.sg (P.C.); marcus.mah@duke-nus.edu.sg (M.G.M.); 2Key Laboratory of Genetic Evolution & Animal Models, Yunnan International Joint Laboratory of Zoonotic Viruses, Yunnan Key Laboratory of Biodiversity Information, Kunming Institute of Zoology, Chinese Academy of Sciences, Kunming 650223, China; yangxinglou@mail.kiz.ac.cn

**Keywords:** hepatitis E virus (HEV), chirohepevirus, bats, genetic diversity, molecular evolution, zoonotic potential

## Abstract

Hepatitis E virus (HEV) is a major zoonotic pathogen causing hepatitis E, with strains identified in various animal species, including pigs, wild boar, rabbits, deer, camels, and rats. These variants are capable of crossing species barriers and infecting humans. HEV belongs to the family *Hepeviridae*, which has recently divided into two subfamilies: *Orthohepevirinae* and *Parahepevirinae*, and five genera: *Paslahepevirus*, *Avihepevirus*, *Rocahepevirus*, *Chirohepevirus*, and *Piscihepevirus*. Recent advances in high-throughput sequencing, particularly of bat viromes, have revealed numerous HEV-related viruses, raising concerns about their zoonotic potential. Bat-derived HEVs have been classified into the genus *Chirohepevirus*, which includes three distinct species. In this study, we analyzed 64 chirohepevirus sequences from 22 bat species across six bat families collected from nine countries. Twelve sequences represent complete or nearly complete viral genomes (>6410 nucleotides) containing the characteristic three HEV open reading frames (ORFs). These strains exhibited high sequence divergence (>25%) within their respective host genera or species. Phylogenetic analyses with maximum likelihood methods identified at least seven distinct subclades within *Chirohepevirus*, each potentially representing an independent species. Additionally, the close phylogenetic relationship between chirohepevirus strains and their bat hosts indicates a pattern of virus–host co-speciation. Our findings expand the known diversity within the family *Hepeviridae* and provide new insights into the evolution of bat-associated HEV. Continued surveillance of chirohepevirus will be essential for understanding its potential for zoonotic transmission and public health risks.

## 1. Introduction

Hepatitis E virus (HEV) is the causative agent of hepatitis E, a viral liver disease that remains one of the leading causes of acute viral hepatitis globally, with at least 20 million HEV infections, an estimated 3.3 million symptomatic cases, and 70,000 deaths occurring annually [1,2]. There are four major HEV genotypes that infect humans: genotypes 1 (HEV-1) and 2 (HEV-2), which are primarily transmitted through the consumption of contaminated water in areas with inadequate sanitation, and genotypes 3 (HEV-3) and 4 (HEV-4), which are mainly zoonotic, transmitted from animals, particularly pigs, and are more prevalent in industrialized countries [3]. In healthy human individuals, HEV infection is typically self-limiting and resolves within a few weeks. However, HEV-1 can lead to severe complications in pregnant women, particularly in the third trimester, with case fatality rates as high as 30% [4]. In contrast, HEV-3 has been associated with chronic infection in immunocompromised individuals, such as organ transplant recipients and patients undergoing chemotherapy, which can result in cirrhosis or even liver failure [5]. However, treatment options for chronic hepatitis E are limited, with ribavirin being the primary antiviral drug [6]. Furthermore, HEV has been implicated in a variety of extrahepatic manifestations, including neurological and renal complications, adding to its clinical complexity [7]. Despite the growing burden of HEV-associated diseases, no specific antiviral treatment for HEV is currently available [8]. As such, HEV continues to pose a significant global health challenge.

HEV is a quasi-enveloped, positive-sense, single-stranded RNA virus with a genome approximately 7.2 to 7.5 kb in length. The viral genome comprises three major open reading frames (ORFs): ORF1, which encodes a non-structural polyprotein involved in virus replication; ORF2, which encodes the viral capsid protein necessary for virus particle formation; and ORF3, which encodes a small phosphoprotein implicated in viral particle release [9]. HEV belongs to the family *Hepeviridae*, which is genetically diverse and currently expanding in taxonomy as increasing new numbers of viruses have been discovered and identified in animal reservoirs [10]. According to the International Committee on the Taxonomy of Viruses (ICTV) in 2022, *Hepeviridae* is divided into two subfamilies: *Orthohepevirinae* and *Parahepevirinae*. The subfamily *Orthohepevirinae* includes four genera (*Paslahepevirus*, *Avihepevirus*, *Rocahepevirus*, and *Chirohepevirus*), which are based on phylogenic analysis and host tropism, while the subfamily *Parahepevirinae* comprises only one genus *Piscihepevirus*, which infects fish such as trout and salmon [11]. The genus *Paslahepevirus* can infect a range of mammalian hosts, including humans, pigs, deer, rabbits, and camels; the genera *Avihepevirus* and *Rocahepevirus* primarily infect birds and rodents, respectively; the genus *Chirohepevirus* was recently identified in bats worldwide [11]. In addition to these well-characterized viruses, there are still many HEV-related strains that remain unclassified due to incomplete genomes or unresolved phylogenetic relationships [10]. Of particular concern is the zoonotic potential of HEV, as variants originating from pigs, wild boar, rabbits, deer, camels, and rats have demonstrated the ability to cross species barriers and infect humans, although the mechanisms underlying zoonotic transmission remain poorly understood [12,13].

Bats, which are known reservoirs for a wide range of viral pathogens, have played a crucial role in the ecology and transmission of emerging infectious diseases [14,15]. Their unique biological traits, including their ability to fly, their migratory behavior, and their distinctive immune system, allows them to harbor a diverse array of viruses, many with zoonotic potential [16]. Bats also live in large colonies and have extensive migratory patterns, facilitating the spread of viruses both within bat populations and between bats and other species, including humans. Their immune systems are particularly noteworthy, as bats can mount an effective antiviral response without inducing the harmful inflammation seen in other species, allowing them to carry viral pathogens without severe diseases [17]. Bats’ exceptional ability to tolerate viral infections has implicated them as reservoirs for a number of high-profile zoonotic viruses, including SARS-CoV-1, SARS-CoV-2, MERS-CoV (family *Coronaviridae*), Ebola and Marburg viruses (family *Filoviridae*), and Hendra and Nipah viruses (family *Paramyxoviridae*) [18,19]. Despite the risks associated with bat-borne diseases, bats play crucial ecological roles, such as controlling insect populations, pollinating plants, and dispersing seeds, which complicates their relationship with human and animal health [20].

HEV-related viruses were first discovered in bats in 2012 and were found to be genetically distinct from human-infecting HEV strains within the genus *Paslahepevirus* [21]. Over the past decade, advances in high-throughput sequencing and metagenomic studies, particularly involving bat viromes, have uncovered a growing number of HEV-related viruses in a variety of bat species. Although these bat-derived viruses are genetically diverse, they share genomic similarities with human-infecting HEV strains [11]. Currently, bat-derived HEV strains are classified within a novel genus, *Chirohepevirus*, which includes three distinct species (*C. desmodi*, *C. eptesici*, and *C. rhinolophi*) based on phylogenetic relationships [10]. While there is currently no direct evidence of chirohepevirus spillover into humans, the genetic similarity between chirohepeviruses and human-infecting HEV strains raises concerns about its zoonotic potential, especially the recent case of zoonotic rat HEV infection [22]. In 2022, we analyzed 18 bat-derived HEV sequences from 12 bat species [11]. Since then, additional chirohepeviruses has been discovered in more bat species. The present study aims to analyze all globally available chirohepevirus sequences to assess the genetic diversity and molecular evolution of these bat-associated HEV strains. By understanding the genetic variability and evolutionary patterns of chirohepeviruses, we aim to better understand the potential risks of zoonotic spillover and expand our knowledge of the broader *Hepeviridae* family.

## 2. Materials and Methods

### 2.1. Dataset Collection

To compile a comprehensive dataset of chirohepevirus sequences, a series of search terms were used to query the NCBI GenBank database up until 15 November 2024. The search terms included “bat hepatitis E virus”, “*bat hepevirus*”, “*chirohepevirus*”, “*chirohepevirus* sp.”, “*chirohepevirus desmodi*”, “*chirohepevirus eptesici*”, “*chirohepevirus rhinolophi*”, and “*unclassified orthohepevirus*”. Additionally, sequences of chirohepevirus were retrieved from the database of bat-associated viruses (DBatVir). A total of 64 sequences were retrieved, including partial genomic sequences and complete or nearly complete genomes of chirohepevirus strains (Table 1).

### 2.2. Sequence Analyses

The genomic sequences of chirohepevirus strains were annotated to indicate the presence of the three typical open reading frames (ORFs) that are characteristic of HEV genomes using Geneious Prime software, version 2025.0.3 (Biomatters Ltd., Auckland, New Zealand). The sequences were then aligned using MAFFT as implemented in Geneious Prime. To assess the genetic variation of chirohepevirus strains, complete genomic sequences were compared with representative strains from the five genera within the family *Hepeviridae*. Pairwise nucleotide distances for the chirohepevirus strains were calculated using Molecular Evolutionary Genetics Analysis (MEGA) software, version 11 [32]. Maximum nucleotide distances were calculated across different host genera and species to evaluate the extent of genetic divergence within the family. The reference sequences for the phylogenetic comparison were selected based on the latest ICTV Virus Taxonomy Profile for *Hepeviridae*. These included the following reference strains: the human HEV prototype Burma strain (GenBank accession no. M73218) for the genus *Paslahepevirus*, the avian HEV prototype F93-5077 strain (GenBank accession no. AY535004) for the genus *Avihepevirus*, the rat HEV prototype R63 strain (GenBank accession no. GU345042) for the genus *Rocahepevirus*, the bat HEV BS7 strain (GenBank accession no. JQ001749) for the genus *Chirohepevirus*, and the fish HEV prototype Heenan Lake strain (GenBank accession no. HQ731075) for the genus *Piscihepevirus* [10].

### 2.3. Phylogenetic and Evolutionary Analyses

Phylogenetic relationships of chirohepevirus strains were reconstructed based on complete genomes using maximum-likelihood (ML) methods in IQ-TREE version 2.3.6, with 1000 ultrafast bootstrap replicates to assess statistical support. The phylogenetic tree was rooted using the piscihepevirus Heenan88 strain from cutthroat trout (GenBank accession no. HQ731075). The phylogenetic inference was visualized using FigTree version 14.4. To specifically examine the phylogenetic relationships within the genus *Chirohepevirus*, separate ML phylogenetic analyses were performed using either partial ORF1 (1600 nucleotides) of 46 chirohepevirus or nearly complete genomes (5600 nucleotides) of 31 chirohepevirus strains. An avian HEV strain (GenBank accession no. AY535004) was used as the outgroup for these analyses to root the trees. Additionally, to specifically investigate the phylogenetic relationships of six HEV-related variants identified in bats in New Zealand, phylogenetic analysis was performed using partial ORF1 (900 nucleotides). A Tuatara cloacal-associated hepevirus strain (GenBank accession no. OP080571) was included for this analysis.

To investigate the potential co-speciation relationship of bat chirohepeviruses and their chiropteran hosts, ML topologies were reconstructed and compared between the partial complete genomes (4400 nucleotides) of 33 chirohepevirus strains and their corresponding 13 bat species, based on mitochondrial cytochrome B (CYTB) gene sequences (1140 nucleotides). The mitochondrial genomes of the bat species were downloaded from the NCBI GenBank database (Appendix A). The host species’ taxonomy was cross-referenced from the NCBI Taxonomy database and Bats of the World: A Taxonomic and Geographic Database version 1.6 [33].

## 3. Results

### 3.1. Detection and Distribution of Chirohepeviruses

A comprehensive search was conducted to identify all publicly available bat-associated HEV and chirohepeviruses from the PubMed, GenBank, and DBatVir databases. In total, 64 chirohepevirus sequences were retrieved, comprising both partial and complete genomic sequences (Table 1). These chirohepeviruses were found in 22 bat species across six bat families: Rhinolophidae, Hipposideridae, Phyllostomidae, Miniopteridae, Mystacinidae, and Vespertilionidae. They are distributed in different geographic regions, including Switzerland, Germany, Sweden, Panama, Peru, Ghana, China, Japan, and New Zealand (Figure 1). The samples were collected between 2008 and 2021, with diverse sample types, including anal swab, feces, blood, liver, bat guano, and pooled animal samples (Table 1).

Notably, our partial RdRp phylogenetic tree revealed a monophyletic group of related HEV strains (shaded in orange in Appendix A) that are closely related to the family *Hepeviridae*. This well-supported clade (ML bootstrap support = 95%) comprises HEV-related strains identified in New Zealand from the lesser short-tailed bat (*Mystacina tuberculata*) and long-tailed bat (*Chalinolobus tuberculatus*). Within this clade, the bat viruses shared >90% amino acid sequence identity with a Tuatara cloacal-associated hepevirus (GenBank accession no. OP080571), detected in cloacal swabs from the tuatara (*Sphenodon punctatus*), a reptile species endemic to New Zealand. Interestingly, one chirohepevirus strain from *M. tuberculata* (7470 nt) (GenBank accession no. OR248815) consists of only one single predicted ORF (2486 aa), albeit lacking the typical three HEV ORFs. This truncated ORF shared the highest amino acid identity of up to 36.8% with the RdRp gene of a Swiper virus (GenBank accession no. QQR34432) identified in red fox (*Vulpes vulpes*) in Australia. Whether this unique sequence represents a spillover from an unknown animal species, given the sample’s origin from bat guano, or a possible sequencing artefact remains unclear. Moreover, this clade is distinct from the family *Hepeviridae*, which consists of a diverse range of HEV strains from various hosts, including bats, humans, swine, rodents, birds, and fish HEV strains (shaded in red in Appendix A).

### 3.2. Genomic Characterization of Bat Chirohepeviruses

While most of the bat chirohepevirus sequences were partial genomic fragments, 12 sequences correspond to complete or nearly complete viral genomes (>6410 nt) (Figure 2 and Table 1). These included strains Rf-HEV/Shanxi/2013 from a Chinese greater horseshoe bat (*Rhinolophus ferrumequinum*), GD2019 and LQB_Rsin from Chinese rufous horseshoe bats (*Rhinolophus sinicus*), API17_F_DrHEV, AYA11_F_DrHE, AYA14_F_DrHEV, and LR3_F_DrHEV from Peruvian common vampire bats (*Desmodus rotundus*), BtHEVMd2350 from a Chinese David’s myotis (*Myotis davidii*), JM_My.ricketti.hev from a Chinese Rickett’s big-footed myotis (*Myotis ricketti*), BS7/GE/2009 from a German common serotine bat (*Eptesicus serotinus*), BtHEV-Ps1/CH/2019 from a Swiss Nathusius’s pipistrelle (*Pipistrellus nathusii*), and JM_Mi.magnater.hev from a Chinese Western long-fingered bat (*Miniopterus magnater*). Notably, the viral genomes of GD2019 and BtHEV-Ps1/CH/2019 lacked the sequences of the 5′ end of the untranslated region (UTR).

In analogy to other mammalian, avian, and fish HEV variants within the family *Hepeviridae*, all bat chirohepeviruses exhibit a similar genomic organization and ORF composition, containing ORF1, ORF2, and ORF3 regions. Excluding the highly divergent piscihepevirus within the *Parahepevirinae* subfamily, some genomic features of bat chirohepeviruses are notably distinct from the three other genera within the subfamily *Orthohepevirinae*. Specifically, chirohepeviruses (~6.5 kb) have a shorter genome length compared to paslahepeviruses (~7.2 kb), avihepevirues (~6.7 kb), and rocahepeviruses (~7.0 kb). Despite their smaller genome size, chirohepeviruses possesses a markedly longer ORF3 (~138 amino acids) compared to paslahepeviruses (~113 amino acids), avihepevirues (~87 amino acids), and rocahepeviruses (~118 amino acids). Additionally, while other HEV strains have distinct translation frames for ORF2 and ORF3, the ORF3 of bat chirohepeviruses is entirely located within the N-terminal region of ORF2. In contrast, the ORF3 of other mammalian and avian orthohepeviruses only partially overlaps with the N-terminal region of ORF2 (Figure 2).

### 3.3. Genetic Diversity of Chirohepeviruses

Maximum likelihood phylogeny based on nearly whole genomes revealed that all bat-derived chirohepeviruses (*n* = 12) formed a well-supported monophyletic clade (BS = 100%), which is distinct from the four other genera *Paslahepevirus*, *Avihepevirus*, *Rocahepevirus*, and *Pischhepevirus* (Figure 3A). All chirohepeviruses exhibited substantial genetic divergence from paslahepeviruses, avihepeviruses, and rocahepeviruses, with pairwise nucleotide sequence distances of 55%, 53%, and 54%, respectively. They were even more distantly related to piscihepevirus, with a pairwise nucleotide sequence distance of up to 67% (Figure 3B). 

Moreover, chirohepeviruses from different bat genera displayed genetic variability, with notable divergence even within the same host species (Figure 3C). The maximum nucleotide sequence distances among horseshoe bats (*Rhinolophus* sp.), common vampire bats (*Desmodus rotundus*), bamboo bats (*Tylonycteris* sp.), and Japanese house bats (*Pipistrellus abramus*) were 32%, 33%, 25%, and 25%, respectively, suggesting long-term evolutionary associations between these HEV viruses and their bat host species [34]. Overall, the nucleotide sequence divergence among all bat-derived chirohepeviruses reached 40%.

### 3.4. Phylogenetic Relationships of Chirohepeviruses

To investigate the phylogenetic relationships among the chirohepeviruses identified in bats, we constructed two maximum likelihood phylogenetic trees: one based on partial ORF1 region (1600 nucleotides; Figure 4) and another using nearly complete genomes (5600 nucleotides, Figure 5). An avian HEV strain (GenBank accession no. AY535004) within the genus *Avihepevirus* was used as an outgroup for these analyses. Among the 46 chirohepevirus strains using partial ORF1 sequences, we identified seven distinct subclades within the genus *Chirohepevirus* (Figure 4). Notably, five of these subclades (5/7) were closely associated with specific bat genera or species, including *Tylonycteris* sp., *Myotis* sp., *Pipistrellus abramus*, *Rhinolophus* sp., and *Desmodus rotundus*. In contrast, one subclade contained chirohepeviruses from several bat genera and species, including *Pipistrellus pygmaeus*, *Pipistrellus nathusii*, *Eptesicus serotinus*, *Eptesicus japonensis*, and *Plecotus sacrimontis*. Another subclade included viruses from *Miniopterus pusillus* and *Scotophilus kuhlii*. Two chirohepevirus strains – one from *Miniopterus magnate* (GenBank accession no. OQ715534) and another from *Myotis ricketti* (GenBank accession no. OQ715533), detected in the same study, clustered together with an exceptionally high nucleotide identity of 99.9% (6556 nt). Whether this chirohepevirus from *Miniopterus magnate* represents a natural infection through a potential cross-species transmission event, or a case of contamination during field sampling or sequencing, remains uncertain. Given these uncertainties, we have cautiously excluded this virus strain from further analysis.

The phylogenetic tree based on nearly complete genome of 31 chirohepevirus strains exhibited a similar topology (Figure 5). According to the latest classification proposed by the ICTV *Hepeviridae* Study Group, three distinct *Chirohepevirus* species have been defined based on virus phylogeny and host tropism: *C. eptesici*, *C. rhinolophi*, and *C. desmodi*, [10]. Specifically, *C. eptesici* includes BS7/GE/2009 from *Eptesicus serotinus* and BtHEVMd2350 from *Myotis davidii*; *C. rhinolophi* includes Rf-HEV/Shanxi2013 from *Rhinolophus ferrumequinum*; and *C. desmodi* includes several strains from *Desmodus rotundus* (API17_F_DrHEV, AYA11_F_DrHE, AYA14_F_DrHEV, and LR3_F_DrHEV) (Figure 2). Based on our phylogenetic inference, here we propose that each subclade may represent a distinct *Chirohepevirus* species. Following ICTV guidelines, which suggest that HEV species names be Latinized descriptors derived from the binomial nomenclature of the virus and its host, we designate the following seven species: *C. tylonicteris* from *Tylonycteris* sp., *C. eptesici* from *Eptesicus* sp., *C. myotis* from *Myotis* sp., *C. pipistrelli* from *Pipistrellus abramus*, *C. miniopteri* from *Miniopterus pusillus* and *Scotophilus khulii*, *C. desmodi* from *Desmodus rotundus*, and *C. rhinolophi* from *Rhinolophus* sp., noting that the virus from *Miniopterus pusillus* was detected earlier than that from *Scotophilus kuhlii*.

### 3.5. Co-Evolution of Chirohepeviruses and Bat Species

To assess the degree of co-evolution between chirohepeviruses and their bat hosts, we performed independent phylogenetic analyses of both the viruses and their host species and examined their co-evolutionary relationships (Figure 6A). The host phylogeny was constructed based on the cytochrome B (CYTB) gene sequences from 13 bat species across four families: Vespertilionidae, Miniopteridae, Phyllostomidae, and Rhinolophidae, that have been identified as hosts for selected chirohepeviruses. The viral phylogeny closely mirrored that of the host phylogeny, suggesting a long history of co-speciation between chirohepeviruses and their bat hosts (Figure 6A). Importantly, although chirohepevirus strains from *Eptesicus serotinus* (Germany) and *Eptesicus japonensis* (Japan) were identified in separate continents, Europe and Asia, respectively, they were grouped together, indicating the wide geographical transmission of their common ancestral chirohepeviruses.

Some chirohepevirus strains display discordant evolutionary patterns compared to their respective bat hosts. For instance, although *Pipistrellus nathusii* and *Pipistrellus abramus* taxonomically belong to the same bat genus, the chirohepevirus detected in *P. nathusii* (Switzerland) clustered with viruses from *Eptesicus* species, forming a distinct group from the strain detected in *P. abramus* (China). Additionally, the chirohepevirus found in *Scotophilus kuhlii* (China) is closely related to a strain from *Miniopterus pusillus* (China), despite the hosts belonging to different families, Miniopteridae and Phyllostomidae. Since both viral strains were identified in the same study and country, it remains unclear whether these bat populations are interconnected and share similar viral strains, especially given the considerable evidence of frequent cross-species transmission events within the *Hepeviridae* family.

While it is evident that chirohepeviruses may have co-evolved with certain bat species, these results must be interpreted with caution. Bats are an extraordinarily diverse group, with more than 1400 species recognized to date, although only a small proportion have been studied. To date, chirohepeviruses have been detected in six bat families (Vespertilionidae, Phyllostomidae, Miniopteridae, Mystacinidae, Hipposideridae, and Rhinolophidae), spanning 22 species. This only represents less than 2% of the 1487 recognized bat species (Figure 6B), highlighting that a large number of bat species remain unexplored for the presence of HEV variants.

## 4. Discussion

Bats are unique among wildlife in their ability to carry viruses without exhibiting symptoms, positioning them as crucial reservoirs for emerging infectious diseases. Bat-associated viruses are particularly concerning due to the vital role bats play as natural hosts for a wide range of pathogens capable of spilling over into humans, such as coronaviruses, Ebola and Marburg viruses, and Hendra and Nipah viruses [17,20,35]. Recent discoveries of HEV-like viruses in multiple bat species worldwide have led to the establishment of a new genus, *Chirohepevirus*, within the family *Hepeviridae* [21,25,30]. This finding significantly broadens our understanding of HEV-related virus diversity, suggesting that bats may harbor a wider range of HEV-like viruses than previously recognized [11].

HEV is distinctive among hepatotropic viruses due to its zoonotic potential, with the ability to be transmitted between humans and animals. HEV ranks sixth in spillover risk among 887 wildlife viruses, underscoring its potential for cross-species transmission [36]. First identified in domestic pigs in the United States in 1997, swine HEV demonstrated zoonotic potential, prompting the identification of various HEV strains closely related to human variants in over a dozen animal species [37]. Among these, *Paslahepevirus balayani* and *Rocahepevirus ratti* species are known to infect humans. Specifically, HEV-1 and HEV-2 (within *P. balayani*) infect only humans, while HEV-3 and HEV-4 can infect both humans and a variety of animal species [12]. Additionally, HEV-5 and HEV-6 have been found in wild boars, and HEV-7 and HEV-8 in camels, with potential implications for human infection [38]. Furthermore, strains of *R. ratti* HEV-C1 in rats have been linked to zoonotic transmission, with over 20 documented cross-species infections globally [39]. The continuous emergence of novel HEV strains in wildlife, particularly those genetically similar to human HEV, highlights the importance of studying viral transmission dynamics and evolution to prevent future outbreaks [13]. Conceivably, the discovery of chirohepevirus strains in bats has raised important public health concerns regarding their zoonotic potential. Our phylogenetic analyses reveal that all known chirohepevirus strains from bats form a distinct monophyletic clade within the family *Hepeviridae*, separate from the clades of HEV variants found in other animal groups. Furthermore, extensive testing of 93,146 plasma samples from blood donors in Germany and 453 serum samples from HIV-infected patients in Cameroon yielded no evidence of bat chirohepevirus RNA, suggesting that these viruses do not currently pose a direct threat to humans [21]. Nevertheless, ongoing surveillance of bat populations and continued research into the zoonotic potential of chirohepeviruses remains important. Understanding the evolutionary processes within bat populations is essential for assessing potential spillover risks, especially as these viruses may evolve over time. 

The discovery of novel HEV-related viruses across various animal species has significantly expanded the taxonomy within the family *Hepeviridae* [40]. Based on our phylogenetic inference and in accordance with the ICTV guidelines, we propose that at least seven distinct species, tentatively named *C. tylonicteris* from *Tylonycteris* sp., *C. eptesici* from *Eptesicus* sp., *C. myotis* from *Myotis* sp., *C. pipistrelli* from *Pipistrellus abramus*, *C. miniopteri* from *Miniopterus pusillus* and *Scotophilus khulii*, *C. desmodi* from *Desmodus rotundus*, and *C. rhinolophi* from *Rhinolophus* sp., could be assigned in the genus *Chirohepevirus*. These species correspond to specific chirohepevirus strains found in different bat species. Although 64 chirohepevirus sequences from six bat families and 22 bat species have been reported, only a few represent full-length or nearly complete viral genomes, with many others consisting of partial fragments. Phylogenetic analyses based on partial viral sequences may affect the robustness of viral evolutionary patterns. Therefore, acquiring more complete chirohepevirus genomes from additional bat species is essential to refining the classification of the genus *Chirohepevirus*. Furthermore, with the growing interest in the global bat virome and advancements in sequencing technologies, we anticipate that novel chirohepeviruses will be discovered in other bat species. 

Nevertheless, our study highlights the substantial genetic diversity within chirohepevirus strains, even within the same host species or genus, suggesting that bats have a long-term association with a diverse pool of genetically related chirohepeviruses. The observed genetic diversity raises the possibility that specific amino acid substitutions may be associated with strain-specific characteristics, such as viral virulence, transmission efficiency, or host specificity, similarly to the observations in human HEV strains [41]. A more detailed comparative analysis of amino acid substitutions across chirohepevirus strains from different host species and geographical regions may help determine whether certain genetic variations correlate with host species or environmental factors [42,43]. Despite the emergence of genetically diverse chirohepevirus strains, there remains a significant gap in research regarding their molecular virology and virus–host interactions. This is largely due to the absence of essential virological tools, such as reverse genetic systems and cell culture models, for these newly identified viruses [44]. Recent studies on paslahepevirus suggest that the papain-like cysteine protease (PCP) domain of ORF1 functions as either a metal-binding domain or a fatty acid binding domain [45,46]. Further structural analysis of HEV, particularly the ORF1 protein of chirohepevirus, would be useful for comparing the binding domain among different genera. Additionally, studying the infectivity and receptor-binding profiles of different chirohepevirus strains requires appropriate animal models. While bats may serve as a natural host for studying these viruses, technical difficulties remain, particularly given the limited understanding of bat immunology and immunopathology. Further research into the functional and structural properties of chirohepeviruses could yield valuable insights into the replication mechanisms and disease processes associated with human HEV.

Co-evolutionary analyses suggests that chirohepevirus strains exhibit a close evolutionary relationship with their chiropteran hosts, indicating virus–host co-speciation. While genetically diverse chirohepevirus strains have been identified in bats, the spillover risks into other mammalian hosts, such as humans, rodents, and pigs, are not known. In contrast, HEV-related viruses recently detected in various rodents may be ancestral to human- and swine-associated HEV variants [47,48,49]. The close phylogenetic relationship between the genera *Paslahepevirus* and *Rocahepevirus* suggests a shared ancestry, providing valuable context for understanding the broader evolutionary landscape of the *Hepeviridae* family [34,50]. As more novel HEV-related viruses continue to be discovered, they are likely to fill our knowledge gaps in understanding HEV evolution and transmission.

In summary, the discovery of chirohepeviruses in bats has significantly expanded the host range and diversity within the *Hepeviridae* family, providing new insights into the molecular characterization and evolutionary origins of HEV across different host species. Further detection and genomic characterization of chirohepevirus variants in additional bat species will be crucial for accurately mapping the evolution of genus *Chirohepevirus* and assessing potential zoonotic risks. Future functional research is essential to fully elucidate the ecology and molecular biology of bat chirohepeviruses.

## Figures and Tables

**Figure 1 viruses-17-00339-f001:**
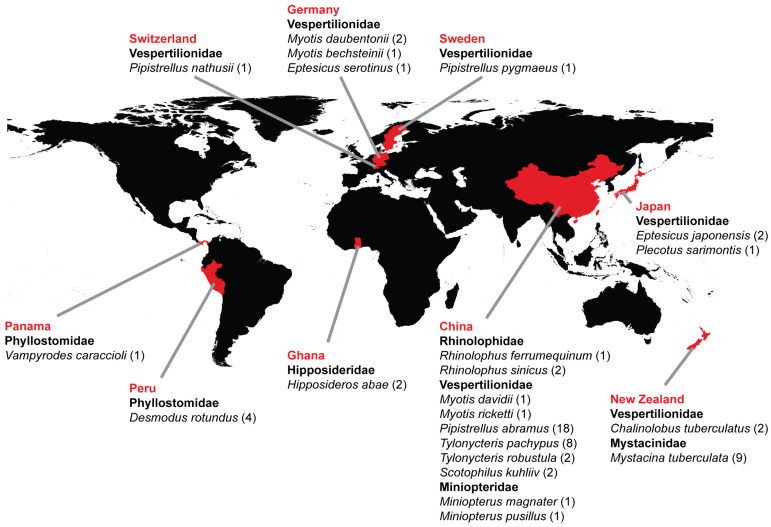
Worldwide distribution of chirohepeviruses in bats. Countries where chirohepevirus genomic sequences have been detected are marked in red. Bat families that are positive for chirohepeviruses are highlighted in bold. The number of chirohepevirus sequences detected in each bat species is shown in brackets. The world map was created using a free and open-source quantum geographic information system (QGIS) version 3.38, with raster map data from Natural Earth.

**Figure 2 viruses-17-00339-f002:**
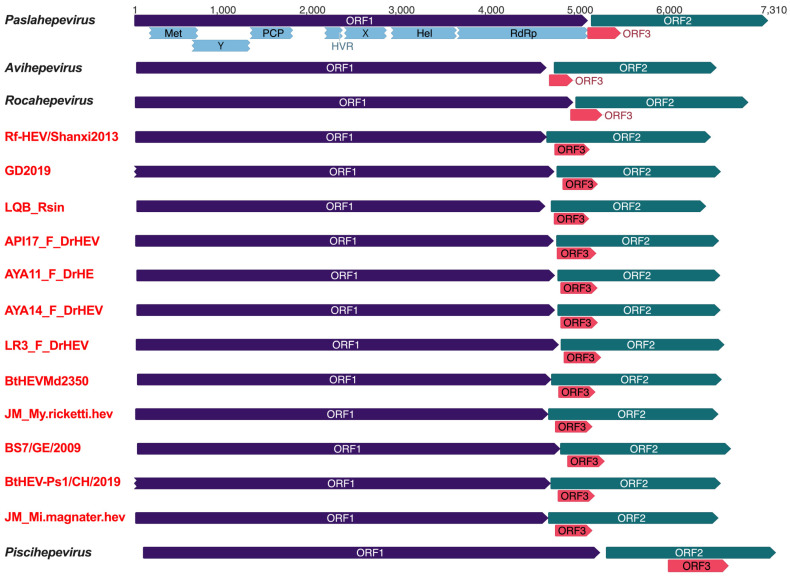
Genome organization comparisons of chirohepeviruses and related genera. The typical three HEV open reading frames (ORFs) and putative functional domains within the ORF1 of *Paslahepevirus* are depicted, including methyltransferase (Met), Y domain, papain-like cysteine protease (PCP), hypervariable region (HVR), X domain, helicase (Hel), and RNA-dependent RNA polymerase (RdRp). The genome scale in nucleotides is shown at the top. GenBank accession numbers for the virus strains presented: M73218 (*Paslahepevirus*), AY535004 (*Avihepevirus*), GU345042 (*Rocahepevirus*), KJ562187 (Rf-HEV/Shanxi2013), MT210622 (GD2019), OR951173 (LQB_Rsin), MW249011 (API17_F_DrHEV), MW249012 (AYA11_F_DrHEV), MW249013 (AYA14_F_DrHEV), MW249014 (LR3_F_DrHEV), KX513953 (BtHEVMd2350), OQ715534 (JM_My.ricketti.hev), JQ001749 (BS7/GE/2009), MT815970 (BtHEV-Ps1/CH/2019), OQ715533 (JM_Mi.magnater.hev), and HQ731075 (*Piscihepevirus*).

**Figure 3 viruses-17-00339-f003:**
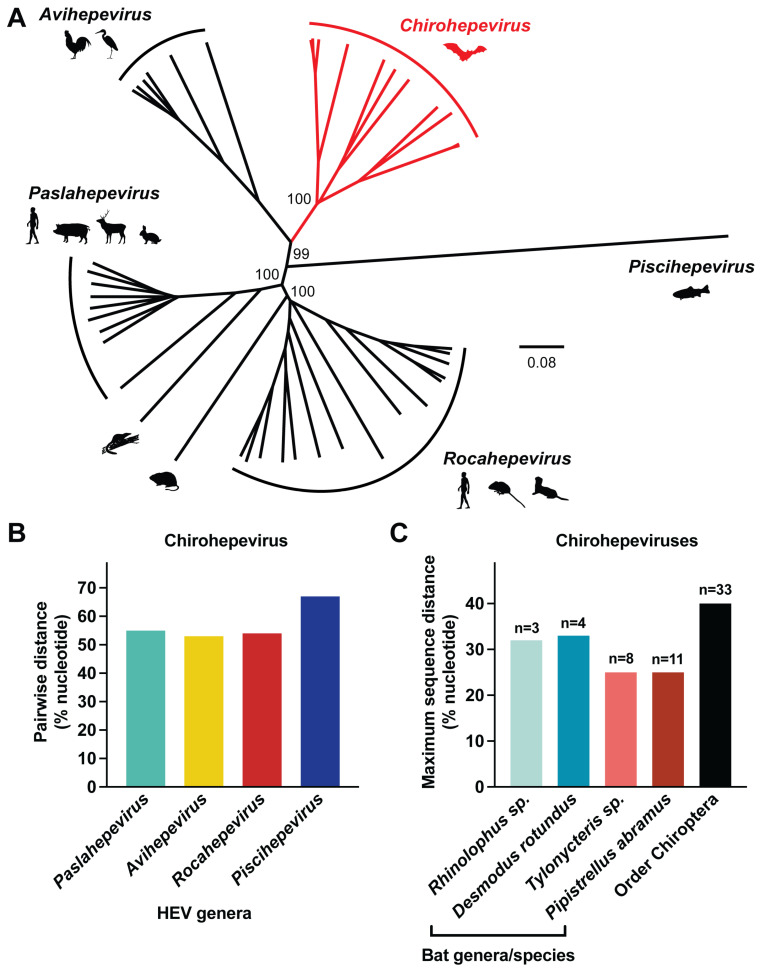
Genetic diversity of chirohepeviruses and other HEV genera. (**A**) Phylogenetic tree based on complete genomes of representative members of the family *Hepeviridae*. Virus classification follows the ICTV Virus Taxonomy Profile: *Hepeviridae* 2022. Virus strains within the genus *Chirohepevirus* are highlighted in red. Major host tropisms for each virus genus are indicated using animal icons. The tree is rooted using the divergent cutthroat trout *Piscihepevirus*. Bootstrap support values are indicated at relevant nodes. The scale bar corresponds to number of nucleotide substitutions per site. (**B**) Pairwise nucleotide sequence distances between *Chirohepevirus* and four other HEV genera. (**C**) Comparison of nucleotide sequence distances based on partial HEV genomes (at least >4400 nucleotides) identified in known bat genera or within the Order Chiroptera.

**Figure 4 viruses-17-00339-f004:**
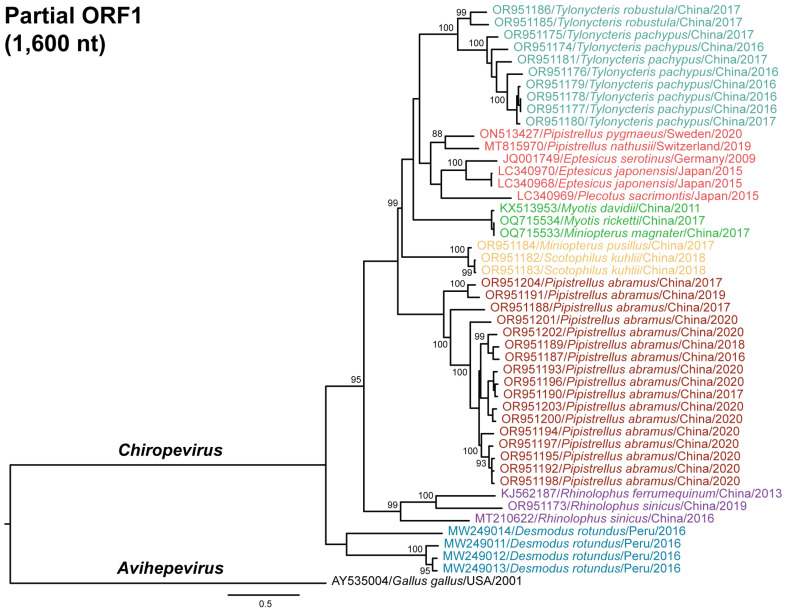
Maximum likelihood phylogeny of partial ORF1 of chirohepeviruses. The tree is based on approximately 1600-nucleotide region of ORF1 and is rooted using an avian HEV strain (GenBank accession no. AY535004) from the genus *Avihepevirus*. Bootstrap support values are indicated at relevant nodes. The scale bar represents the number of nucleotide substitutions per site. Viruses from distinct subclades are color-coded, with designations including GenBank accession number, host species, and sampling location and year.

**Figure 5 viruses-17-00339-f005:**
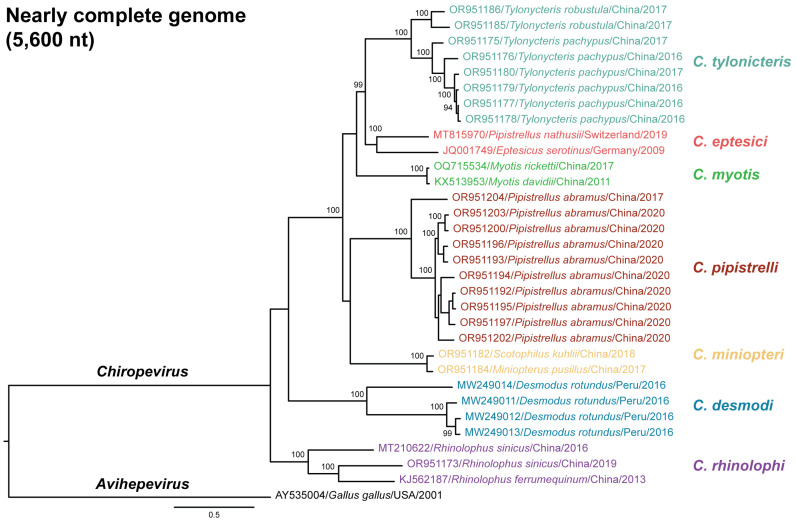
Maximum likelihood phylogeny of nearly complete genomes of chirohepeviruses. The nearly complete genomes consist of approximately 5600 nucleotides. The tree is rooted using an avian HEV strain (GenBank accession no. AY535004) from the related genus *Avihepevirus*. Bootstrap support values are indicated at relevant nodes. The scale bar corresponds to number of nucleotide substitutions per site. Viruses from distinct subclades are highlighted in various colors. Virus designations include GenBank accession number, host species, sampling location, and collection year. Seven potentially designated bat *Chirohepevirus* species are shown on the right.

**Figure 6 viruses-17-00339-f006:**
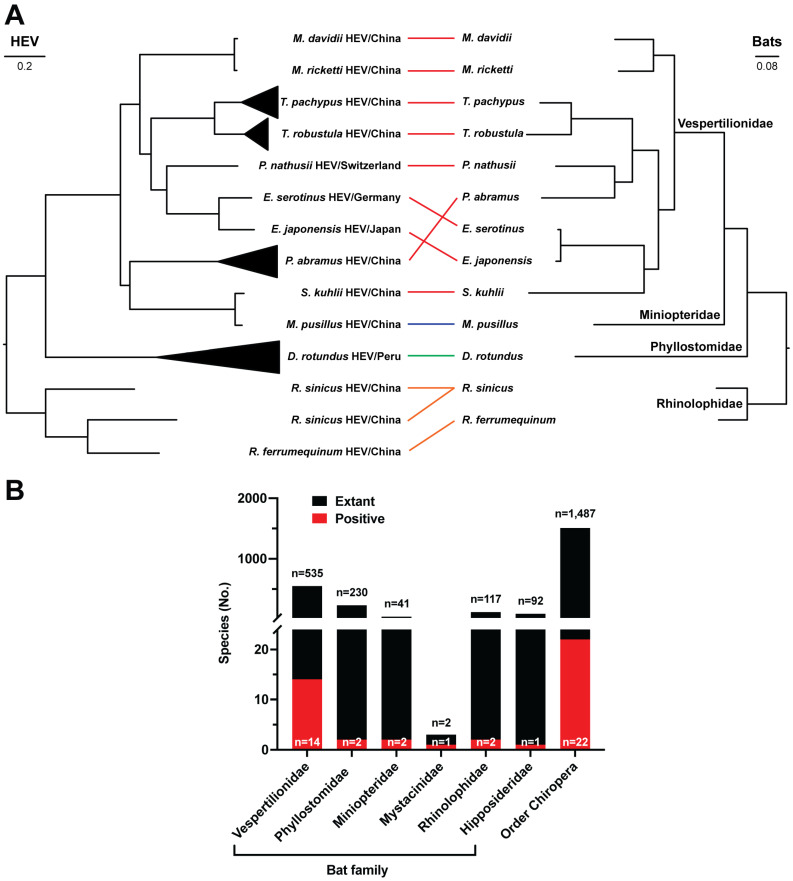
Co-evolution of chirohepeviruses and bat species. (**A**) Maximum likelihood trees generated based on approximately 4400 nucleotide sequences of chirohepeviruses in various bat species (left) and nucleotide sequences of the cytochrome B (CYTB) gene of corresponding hosts (right). Host taxonomy (family) is specified on the phylogenetic tree. Lines between chirohepeviruses and their corresponding host species are color-coded according to the four bat families. The scale bar corresponds to the number of nucleotide substitutions per site. (**B**) Comparison of the numbers of bat species testing positive for chirohepevirus versus the total number of extant bat species in each family or within the Order Chiroptera. The numbers of taxonomically described chiropteran species in each family are derived from the Bat Species of the World databases.

**Table 1 viruses-17-00339-t001:** Detection of chirohepeviruses in diverse bat species.

Host Family	Host Species	Host Common Name *^a^*	Sampling Country (Collection Year)	Sample Source	Genomic Sequence (No.)	GenBank Accession No.	Reference
Rhinolophidae	*Rhinolophus ferrumequinum*	greater horseshoe bat	China (2013)	Anal swab	Complete (1)	KJ562187	[23]
Rhinolophidae	*Rhinolophus sinicus*	Chinese rufous horseshoe bat	China (2016, 2019)	Anal swab	Complete (2)	MT210622, OR951173	N.A. *^b^*, [24]
Hipposideridae	*Hipposideros abae*	Aba roundleaf bat	Ghana (2009)	Feces	Partial (2)	JQ001744, JQ071861	[21]
Phyllostomidae	*Vampyrodes caraccioli*	great stripe-faced bat	Panama (2009)	Blood	Partial (1)	JQ001745	[21]
Phyllostomidae	*Desmodus rotundus*	common vampire bat	Peru (2016)	Anal swab	Complete (4)	MW249011–MW249014	[25]
Miniopteridae	*Miniopterus magnater*	Western long-fingered bat	China (2017)	Pooled animal	Complete (1)	OQ715533	[26]
Miniopteridae	*Miniopterus pusillus*	small bent-winged bat	China (2017)	Anal swab	Partial (1)	OR951184	[24]
Mystacinidae	*Mystacina tuberculata*	New Zealand lesser short-tailed bat	New Zealand (2013, 2020, 2021)	Bat guano	Partial (9)	KM204384, KM204385 *^c^*, OR248814–OR248820	[27,28]
Vespertilionidae	*Chalinolobus tuberculatus*	New Zealand long-tailed bat	New Zealand (2020)	Bat guano	Partial (2)	OR248813, OR248821	[28]
Vespertilionidae	*Myotis daubentonii*	Daubenton’s bat	Germany (2009)	Feces	Partial (2)	JQ001746, JQ001747	[21]
Vespertilionidae	*Myotis bechsteinii*	Bechstein’s bat	Germany (2008)	Feces	Partial (1)	JQ001748	[21]
Vespertilionidae	*Myotis davidii*	David’s myotis	China (2011)	Liver	Complete (1)	KX513953	[29]
Vespertilionidae	*Myotis ricketti*	Rickett’s big-footed myotis	China (2017)	Pooled animal	Complete (1)	OQ715534	[26]
Vespertilionidae	*Eptesicus serotinus*	common serotine bat	Germany (2009)	Liver	Complete (1)	JQ001749	[21]
Vespertilionidae	*Eptesicus japonensis*	Japanese short-tailed bat	Japan (2015)	Feces	Partial (2)	LC340968, LC340970	[30]
Vespertilionidae	*Pipistrellus nathusii*	Nathusius’s pipistrelle	Switzerland (2019)	Feces	Nearly complete (1) *^d^*	MT815970	[31]
Vespertilionidae	*Pipistrellus pygmaeus*	soprano pipistrelle	Sweden (2020)	Feces	Partial (1)	ON513427	N.A.
Vespertilionidae	*Pipistrellus abramus*	Japanese house bat	China (2016–2020)	Anal swab	Partial (18)	OR951187–OR951204	[24]
Vespertilionidae	*Tylonycteris pachypus*	lesser bamboo bat	China (2016–2017)	Anal swab	Partial (8)	OR951174–OR951181	[24]
Vespertilionidae	*Tylonycteris robustula*	greater bamboo bat	China (2017)	Anal swab	Partial (2)	OR951185–OR951186	[24]
Vespertilionidae	*Plecotus sacrimontis*	Japanese long-eared bat	Japan (2015)	Feces	Partial (1)	LC340969	[30]
Vespertilionidae	*Scotophilus kuhlii*	lesser Asiatic yellow house bat	China (2018)	Anal swab	Partial (2)	OR951182–OR951183	[24]
Total	22 species				64 sequences		

*^a^* Host common name is according to the GenBank Taxonomy Browser (https://www.ncbi.nlm.nih.gov/taxonomy/, accessed on 15 November 2024); *^b^* N.A. denotes not available. These GenBank accession numbers do not yet appear in any publication which reports or discusses these data; *^c^* These two viral fragments derived from a single sample with different sequencing method; *^d^* Viral genome lacks 5’ end.

## Data Availability

The nucleotide sequences from all chirohepeviruses analyses in this study are derived from the NCBI GenBank database (https://www.ncbi.nlm.nih.gov, accessed on 15 November 2024).

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
