# Peer review of "Genetic Diversity and Molecular Evolution of Hepatitis E Virus Within the Genus *Chirohepevirus* in Bats"

_viruses, 2025, doi:10.3390/v17030339_

Round 1
Reviewer 1 Report
Comments and Suggestions for Authors
Minor revisions:
LINE 131: ‘…representative strains from the FIVE genera within…’
LINE 251: (Pipistrellus abramus were 32%, … CLOSING BRACKET IS MISSING
LINE 254: FIGURE 3C: THE MEANING OF THE BLACK HISTOGRAM IS CONFUSING. WHAT DOES IT MEAN “all bats”? ARE THE REMAINING SPECIES EXCEPT FOR THOSE INDICATED IN FIG. 3C? WHY n=33? THE SUM OF SEQUENCES SHOLULD BE 64 AND NOT 59. EXPLAIN BETTER (partial ORF1 region or nearly complete genomes?) OR CORRECT DATA.
LINE 299-307 AND Figure 5: THE USE OF LATIN LANGUAGE IS NOT ADEQUATELY USED FOR ALL HEV STRAINS. IT SHOULD BE C. tylonicteris, C. eptesici, C. myotis, C. pipistrelli, C. miniopteri, C. desmodi, C. rinolophi. SEE ALSO DISCUSSION
LINE 310: EDIT “SISTER” TO “RELATED”
LINE 347: Figure 6B: AGAIN, IT IS NOT CLEAR WHY n=1487 IS NOT THE SUM OF THE n EXTANT SPECIES IN THE FIGURE
LINE 353: Comparison
Author Response
LINE 131: ‘…representative strains from the FIVE genera within…’
Reply: We thank the reviewer for this comment. We have changed “four” to “five” (line: 133).
LINE 251: (Pipistrellus abramus were 32%, … CLOSING BRACKET IS MISSING
Reply: We have added the closing bracket (line: 257).
LINE 254: FIGURE 3C: THE MEANING OF THE BLACK HISTOGRAM IS CONFUSING. WHAT DOES IT MEAN “all bats”? ARE THE REMAINING SPECIES EXCEPT FOR THOSE INDICATED IN FIG. 3C? WHY n=33? THE SUM OF SEQUENCES SHOLULD BE 64 AND NOT 59. EXPLAIN BETTER (partial ORF1 region or nearly complete genomes?) OR CORRECT DATA.
Reply: We apologized for the confusion. The black histogram represents the total number of HEV sequences identified from the Order Chiroptera (n=33) including the four listed species. These sequences correspond to partial genomes that are at least more than 4,400 nucleotides in length. We have modified the text in Figure 3B and added the relevant details in figure legend accordingly (revised Figure 3B, lines 268-269).
LINE 299-307 AND Figure 5: THE USE OF LATIN LANGUAGE IS NOT ADEQUATELY USED FOR ALL HEV STRAINS. IT SHOULD BE C. tylonicteris, C. eptesici, C. myotis, C. pipistrelli, C. miniopteri, C. desmodi, C. rinolophi. SEE ALSO DISCUSSION
Reply: We agreed and now consistently use the Latin names of proposed bat HEV species throughout the revised manuscript (lines 308-311, 402-405) and Figure 5B.
LINE 310: EDIT “SISTER” TO “RELATED”
Reply: Edited as request (line 316).
LINE 347: Figure 6B: AGAIN, IT IS NOT CLEAR WHY n=1487 IS NOT THE SUM OF THE n EXTANT SPECIES IN THE FIGURE
Reply: We apologized for the oversight. We have now revised the total number of HEV sequences from the “Order Chiroptera” (revised Figure 6B). Also, we have added the relevant information in the legend for Figure 6B (line 361).
LINE 353: Comparison
Reply: We have replaced “Comparisons” with “Comparison” (line 359).

Reviewer 2 Report
Comments and Suggestions for Authors
This manuscript analysed chirohepevirus sequences available in NCBI Genbank database until November 15, 2024. A total of 64 chirohepevirus sequences including 12 nearly complete viral genomes from 22 bat species collected from nine countries were analysed. This topic is interesting because bat-derived chirohepeviruses share genomic similarities with human-infecting Paslahepevirus and Rocahepevirus. The genetic diversity and evolutionary patterns of chirohepeviruses could help to understand the potential risks of zoonotic spillover. The manuscript is well written.
Comments
- A major limitation of this manuscript is the lack of novelty since the previous paper published by the first author in Viruses in 2022 (reference 14). The added value of the present manuscript in the context of the review published 3 years ago must be explained.
- Introduction
Ribavirin : “…its efficacy remains controversial”. This is not supported by all published data indicating antiviral activity against HEV including references 10 and 11.
- Results, lines 192-193 : a more precise description of OR248815 sequence is needed. It is mentioned that the typical three ORF are lacking. What is the genetic organization of this strain ? Is there a possibility of sequencing artefact ?
- Recent data on Paslahepevirus (Goulet Viruses 2022; Fieulaine Virology 2023) showed that the PCP domain of ORF1 is in fact a metal binding domain or a fatty acid binding domain. It could be interesting to perform a similar structural analysis for Chirohepevirus
- The length of ORF3 is not 123 aminoacids for paslahepeviruses. It is 113 aminoacids for HEV-3, 7 and 8, 114 aminoacids for HEV-1, 2 and 4 and 112 aminoacids for HEV-5 and 6.
Author Response
This manuscript analysed chirohepevirus sequences available in NCBI Genbank database until November 15, 2024. A total of 64 chirohepevirus sequences including 12 nearly complete viral genomes from 22 bat species collected from nine countries were analysed. This topic is interesting because bat-derived chirohepeviruses share genomic similarities with human-infecting Paslahepevirus and Rocahepevirus. The genetic diversity and evolutionary patterns of chirohepeviruses could help to understand the potential risks of zoonotic spillover. The manuscript is well written.
Reply: We thank the expert reviewer for the positive comments and recognition of our manuscript.
A major limitation of this manuscript is the lack of novelty since the previous paper published by the first author in Viruses in 2022 (reference 14). The added value of the present manuscript in the context of the review published 3 years ago must be explained.
Reply: We thank the reviewer for this comment. Indeed, we published a review paper in 2022 (PMID: 35632647) focusing on the topic of bat HEV. However, in the past three years, additional bat HEV-related viruses have been discovered in 10 more bat species, and the number of bat HEV-related sequences has significantly increased (64 compared to 18). Consequently, the availability of a larger dataset of bat HEV sequences greatly enhances the genetic and evolutionary analysis of these viruses within the Chirohepevirusgenus. In response to the reviewer’s suggestion, we have added and explained the value of the present manuscript in the context of the review published three years ago: “In 2022, we analyzed 18 bat-derived HEV sequences from 12 bat species [11]. Since then, additional chirohepeviruses has been discovered in more bat species. The present study aims to analyze all globally available chirohepevirus sequences to assess the genetic diversity and molecular evolution of these bat-associated HEV strains” (lines 103-107).
Introduction
Ribavirin: “…its efficacy remains controversial”. This is not supported by all published data indicating antiviral activity against HEV including references 10 and 11.
Reply: We agree. We have rephrased our statement: “However, treatment options for chronic hepatitis E are limited, with ribavirin being the primary antiviral drug” (lines 47-49).
Results, lines 192-193: a more precise description of OR248815 sequence is needed. It is mentioned that the typical three ORF are lacking. What is the genetic organization of this strain? Is there a?
Reply: As suggested by the reviewer, we have added a more precise description of the sequence (GenBank accession no. OR248815): “Interestingly, one chirohepevirus sequence from M. tuberculata (7,470 nt) (GenBank ac-cession no. OR248815) only consisted of one single predicted ORF (2,486 aa) albeit lacked the typical three HEV ORFs. This truncated ORF shared the highest amino acid identity of up to 36.8% with the RdRp gene of a Swiper virus (GenBank accession no. QQR34432) identified in red fox (Vulpes vulpes) in Australia. Whether this unique sequence represents a spillover from an unknown animal species, given the sample’s origin from bat guano, or a possible sequencing artefact remains unclear” (lines 194-201).
Recent data on Paslahepevirus (Goulet Viruses 2022; Fieulaine Virology 2023) showed that the PCP domain of ORF1 is in fact a metal binding domain or a fatty acid binding domain. It could be interesting to perform a similar structural analysis for Chirohepevirus
Reply: We thank the reviewer for this insightful comment. We noted that these two recent papers (PMIDs: 35891516 and 36527931) used in silico approach to predict the ORF1 protein of human-infecting HEV-3. It would be indeed interestingly to predict the ORF1 of chirohepevirus and perform a comparative analysis with other HEV variants. However, our study primarily focused on the genetic diversity and molecular evolution of chirohepevirus, but structural analysis is beyond the scope of this study. We have added this point to the revised discussion: “Recent study on paslahepevirus suggest that the papain-like cysteine protease (PCP) domain of ORF1 functions as either a metal binding domain or a fatty acid binding domain [45, 46]. Further structural analysis of HEV particularly the ORF1 protein of chirohepevirus would be useful for comparing the binding domain among different genera” (lines 427-431).
The length of ORF3 is not 123 amino acids for paslahepeviruses. It is 113 amino acids for HEV-3, 7 and 8, 114 amino acids for HEV-1, 2 and 4 and 112 amino acids for HEV-5 and 6.
Reply: We thank the reviewer for pointing this out. We have revised our text accordingly (lines 238-239).

Round 2
Reviewer 2 Report
Comments and Suggestions for Authors
The answers are satisfactory